# Associations Between Metabolic Age, Sociodemographic Variables, and Lifestyle Factors in Spanish Workers

**DOI:** 10.3390/nu16234207

**Published:** 2024-12-05

**Authors:** Ignacio Ramírez-Gallegos, Marta Marina-Arroyo, Ángel Arturo López-González, Daniela Vallejos, Emilio Martínez-Almoyna-Rifá, Pedro Juan Tárraga López, José Ignacio Ramírez-Manent

**Affiliations:** 1ADEMA-Health Group University Institute of Health Sciences Research (IUNICS), 07009 Palma, Balearic Islands, Spain; ignacioramirezgallegos@gmail.com (I.R.-G.); marinaarroyomarta@gmail.com (M.M.-A.); d.vallejos@eua.edu.es (D.V.); emilio@udemax.com (E.M.-A.-R.); joseignacio.ramirez@ibsalut.es (J.I.R.-M.); 2Faculty of Dentistry, University School ADEMA, 07009 Palma, Balearic Islands, Spain; 3Balearic Islands Institute of Health Research (IDISBA), Balearic Islands Health Research Institute Foundation, 07010 Palma, Balearic Islands, Spain; 4Balearic Islands Health Service, 07010 Palma, Balearic Islands, Spain; 5Faculty of Medicine, University of Castilla la Mancha, 02008 Albacete, Castilla-La Mancha, Spain; pjtarraga@sescam.jccm.es; 6SESCAM (Servicio Salud Castilla La Mancha), 45071 Toledo, Castilla-La Mancha, Spain; 7Faculty of Medicine, University of the Balearic Islands, 07010 Palma, Balearic Islands, Spain

**Keywords:** metabolic age, mediterranean diet, physical activity, smoking, social class, lifestyle

## Abstract

Background: Metabolic age is defined as an estimation of a person’s age based on their basal metabolic rate (BMR) and other physiological health indicators. Unlike chronological age, which simply measures the number of years lived since birth, metabolic age is based on various health and fitness markers that estimate the body’s “true” biological age and can be assessed using various methodologies, including bioimpedance. The aim of this study was to evaluate how age, sex, social class, smoking habits, physical activity, and adherence to the Mediterranean diet influence metabolic age. Methods: A cross-sectional, descriptive study was conducted on 8590 Spanish workers in the Balearic Islands. A series of sociodemographic variables and health-related habits were assessed, while metabolic age was measured using bioimpedance. A metabolic age exceeding chronological age by 12 years or more was considered high. A descriptive analysis of categorical variables was performed by calculating their frequency and distribution. By applying multivariate models, specifically multinomial logistic regression, we observe that all independent variables (sex, age, social class, physical activity, mediterranean diet, and smoking) show varying levels of association with the occurrence of high metabolic age values. Among these independent variables, those showing the highest degree of association, represented by odds ratios, are physical activity, adherence to the Mediterranean diet, and social class. In all cases, the observed differences demonstrate a high level of statistical significance (*p* < 0.001). Results: The factors with the greatest influence were physical inactivity, with an OR of 5.07; and low adherence to the Mediterranean diet, with an OR of 2.8; followed by social class, with an OR of 2.51. Metabolic age increased with chronological age and was higher in males, with an OR of 1.38. Smoking also had a negative impact on metabolic age, with an OR of 1.19. Conclusions: Mediterranean diet is associated with a higher metabolic age. The most influential factors on metabolic age are physical activity and adherence to the Mediterranean diet, followed by the individual’s socioeconomic class. Smoking also contributes to increased metabolic age, albeit to a lesser extent.

## 1. Introduction

Metabolic age is a relatively new concept in the field of health and wellness. It is defined as an estimate of a person’s age based on their basal metabolic rate (BMR) and other indicators of physiological health, providing an assessment of the body’s ‘true’ biological age [1,2].

Metabolic age is a key indicator of physical fitness, and a lower metabolic age is associated with better physical fitness and a longer, healthier life. Metabolic age is influenced by factors such as genetics, body composition, physical activity, and diet. It alters BMR, which is affected by muscle mass [3], genetic factors [4,5], sex [6], age [7,8], chronic diseases [9], diet [10,11,12], stress [13,14], and sleep quality [15,16]. It has significant implications for both individual health and healthcare costs. When a person’s metabolic age exceeds their chronological age, potential years of health may be lost. The concept of Avoidable Lost Life Years (ALLY) is introduced as a useful measure to assess this loss.

Interventions to improve metabolic age include lifestyle changes, such as increased physical activity, a balanced diet, and stress management [17,18]. Resistance training can be particularly effective for increasing muscle mass and improving BMR [19]. Additionally, nutritional interventions that promote body fat loss and metabolic health improvement are also recommended [20].

Metabolic age can be assessed using various methodologies. One of the most common is the use of bioimpedance scales which estimate body composition and BMR [21]. These scales send a low-intensity electrical current through the body to measure the resistance of different tissues [22].

Another tool used is indirect calorimetry, which measures oxygen consumption and carbon dioxide production to calculate BMR [23,24]. Blood tests evaluating metabolic markers, such as glucose and lipid levels, can also provide valuable information about metabolic health [25].

Having a metabolic age significantly higher than the chronological age can be an indicator of a higher risk of chronic diseases, such as type 2 diabetes, cardiovascular diseases, and certain types of cancer [26]. On the other hand, a metabolic age lower than the chronological age can be a sign of good health and a lower risk of these diseases [27].

Identifying the variables that most influence the metabolic age of our population can be highly valuable for implementing health interventions aimed at reducing metabolic age, improving quality of life within this population, and lowering healthcare costs.

The objective of this study was to evaluate how different variables, both sociodemographic (age, sex, and social class) and health-related habits (smoking, physical activity, and Mediterranean diet), affect metabolic age values determined by bioimpedance in a group of workers in the Balearic Islands.

## 2. Materials and Methods

### 2.1. Participants

A cross-sectional, descriptive study was conducted on 8590 Spanish workers in the Balearic Islands. The sample consisted of workers who attended their annual occupational health check-up between January 2019 and December 2020 at an occupational health and risk prevention service. This service caters to a range of companies, notably in the healthcare, public administration, hospitality, retail, transportation, education, industrial, and cleaning sectors.

Refer to the flow chart in Figure 1 for details.

Inclusion criteria:Individuals aged 18 to 69 years.Willingness to participate in the research.Consent for the data to be used in epidemiological studies.Employment with one of the companies participating in the research and not being temporarily disabled at the time of the study.Exclusion criteria: Age under 18 years or over 69 years.Not being an employee of one of the participating companies.Refusal to participate in the research study.Refusal to consent to the use of data for epidemiological studies.Lacked a parameter for calculating scales.


### 2.2. Determination of Variables

The health personnel from the occupational health services of the participating companies were responsible for gathering the necessary data for this study through:

Anamnesis: A detailed clinical history was compiled, which included information on sociodemographic variables such as age, sex, social class, tobacco consumption, physical exercise, and adherence to the Mediterranean diet.

Anthropometric and clinical measurements: These included height, weight, waist and hip circumference, and systolic and diastolic blood pressure.

Analytical determinations: The lipid profile and blood glucose levels were measured.

#### 2.2.1. Anthropometric Determinations

To minimize potential biases in the study, measurement techniques for the variables were standardized. Height and weight were measured using a SECA 700 scale and a SECA 220 stadiometer. Measurements were taken with the individual in underwear, following international standards for anthropometric evaluation as outlined by ISAK [28]. Data were recorded in centimeters and kilograms.

For the assessment of abdominal waist circumference, a SECA measuring tape was used, placed midway between the last rib and the iliac crest, parallel to the floor. The individual stood with their abdomen relaxed. Hip circumference was measured similarly, with the tape measure parallel to the floor at the widest part of the buttocks [29].

#### 2.2.2. Clinical Determinations

Blood pressure was measured with an OMRON-M3 model blood pressure monitor. For accurate assessment, the individual sat with their back supported by the chair and rested for at least 10 min. The patient needed to be relaxed, with their arm supported at heart level, without crossing their legs. They should not have eaten, smoked, or consumed alcohol or tea for at least an hour before the measurement. The cuff was placed around the upper arm, 2–3 cm above the elbow crease, ensuring a snug fit without being too tight. Various cuff sizes were available. Three consecutive measurements were taken at one-minute intervals, with the final value being the average of the three.

#### 2.2.3. Analytical Determinations

Blood samples were collected via venipuncture after a 12 h fast. Samples were then processed and refrigerated to ensure proper preservation for no more than 48–72 h. Analysis of the samples was conducted in reference laboratories using standardized methodologies. Triglycerides, total cholesterol, and blood glucose were measured using enzymatic methods, while HDL cholesterol was assessed using precipitation methods. LDL cholesterol was calculated indirectly using the Friedewald formula, valid when triglyceride levels did not exceed 400 mg/dL. If triglyceride levels were greater than 400 mg/dL, LDL was measured directly. All analytical variables are expressed in mg/dL.

#### 2.2.4. Risk Scales

Adherence to the Mediterranean diet was evaluated using the PREDIMED questionnaire [30], which consists of 14 questions, each scoring 0 or 1. A score of nine or more indicates high adherence to the diet [31].

The level of physical activity was measured using the International Physical Activity Questionnaire (IPAQ), a self-administered survey that evaluates physical activity over the previous seven days [32].

A person who had smoked at least one cigarette per day (or its equivalent in other forms of consumption) in the past thirty days, or who had quit smoking less than twelve months before, was considered a smoker. A person who had not smoked in the past year or who had never smoked was considered a non-smoker.

Determination of socioeconomic class followed the recommendation of the Spanish Society of Epidemiology, based on the 2011 National Classification of Occupations [33]. According to this classification: Class I includes managers, directors, and university professionals; Class II comprises intermediate professions and self-employed individuals; and Class III consists of manual laborers.

Metabolic age was measured using a TANITA MC-780 S MA bioimpedance meter (TANITA Corporation, Tokyo, Japan).

ALLY (Avoidable Lost Life Years) was calculated by subtracting metabolic age from biological age. Some published studies have reported that a difference of at least 12 years between chronological and metabolic age reduces cardiovascular risk. ALLY is classified as low if the difference is less than 3 years, normal if it is between 3 and 11 years, and high if the difference is 12 years or more. A metabolic age 12 years or more above the chronological age was considered high [34]. This serves as the cutoff point for establishing high metabolic age values for ALLY.

### 2.3. Statistical Analysis

A descriptive analysis of categorical variables was performed, calculating their frequency and distribution. For quantitative variables with a normal distribution, the mean and standard deviation were calculated. To compare means, Student’s *t*-test was used, and to compare proportions, the chi-square test was used. The dependent variable, ALLY, was classified into three categories. Since it comprises more than two categories, we performed a multivariate analysis stratified by age groups to assess the influence of each studied variable within each stratum. The independent variables selected were those considered, according to the reviewed literature, to be the most statistically and biologically recommended. The multivariate analysis was conducted using multinomial logistic regression, along with the calculation of odds ratios and the Hosmer–Lemeshow goodness-of-fit test. SPSS 29.0 software was used for the statistical analysis, with an accepted level of statistical significance of 0.05.

## 3. Results

The anthropometric and clinical details of the study participants are presented in Table 1. The analysis included a total of 8590 individuals (4104 men, 47.8%, and 4486 women, 52.2%). The average age of the sample was slightly over 41 years, with the majority of participants between 30 and 49 years old. The analysis of anthropometric, clinical, and analytical variables revealed significantly lower values in females across all measures. The majority of the sample subjects belonged to social class I. In both sexes, just over 15% were smokers. Among men and women, 25.9% and 35.1%, respectively, did not engage in regular physical exercise. Additionally, more than half of the sample in both sexes adhered to the Mediterranean diet.

Table 2 presents the mean values of the ALLY metabolic age, stratified by age and various variables for each sex. Our results include several negative values, indicating that the metabolic age is lower than chronological age, particularly noticeable in the youngest age group, 18 to 29 years. In nearly all values obtained, the mean metabolic age is lower in women compared to men. In both sexes, the ALLY metabolic age averages worsen with decreasing socioeconomic status, smoking, lack of regular physical activity, and low adherence to the Mediterranean diet. Notably, regular physical activity, practiced at least three days a week, emerges as the variable most significantly associated with a lower metabolic age across all ages and in both sexes. In all the results, the observed differences are statistically significant (*p* < 0.001).

Table 3 presents the prevalence of high ALLY metabolic age values. When comparing between sexes, the results consistently show a lower percentage in women, suggesting that the studied variables may have less influence on the ALLY metabolic age in females. Comparing across age groups reveals an increase in the percentage of individuals with high metabolic age values with advancing age, reflecting trends related to socioeconomic status and unhealthy lifestyle habits. This pattern is consistent across both sexes. Interestingly, a similar trend is observed for variables associated with healthy lifestyle behaviors (e.g., non-smoking, engaging in regular physical activity three or more days per week, and adherence to the Mediterranean diet). This may indicate that the influence of these factors on metabolic age diminishes with age or that their effects are confounded by interactions between variables. In all analyzed outcomes, the differences observed were statistically significant (*p* < 0.001).

We performed a multinomial logistic regression analysis stratified by age groups to assess the effect of sex, social class, physical activity, adherence to the Mediterranean diet, and tobacco consumption on the likelihood of presenting with a high ALLY metabolic age (dependent variable), as shown in Table 4. This approach allowed us to evaluate the risk associated with each variable independently. Across all age groups, we observed gender differences, with a higher risk of high ALLY metabolic age in men compared to women. Although all modifiable variables showed an increased risk for high ALLY metabolic age, the most influential factors (highest odds ratios) were physical activity and adherence to the Mediterranean diet. Therefore, it is crucial to focus efforts on improving adherence to these two factors in primary care consultations, without neglecting the others. In all the factors studied, the observed differences were statistically significant (*p* < 0.001).

## 4. Discussion

Biological aging and chronological age are closely related but not synonymous. Advanced age is associated with decreased functional capacity, a chronic hyperinflammatory state, and a higher prevalence of chronic diseases, which may accelerate metabolic age [35]. Previous studies have shown that advanced chronological age is associated with a higher risk of metabolic diseases and increasing metabolic age [36]. Moreover, deterioration of the endocrine and immune systems, along with decreased muscle mass and increased fat mass, are characteristics of aging that contribute to a higher metabolic age. However, biological aging is not a disease but rather an ascending process that occurs progressively in all living beings that surpass the so-called essential life span (ELS) of their species. For humans, this ELS point is typically between 40 and 50 years of age. An individual’s ability to survive beyond this point serves as an indicator of their health status, their capacity to endure, and ability to maintain a healthy functional state. Therefore, it is crucial to identify the factors that influence an individual’s metabolic age in order to intervene in their homeodynamics and enhance their functional status and quality of life [37].

In our sample, metabolic age increased with chronological age, lower socioeconomic status, smoking, sedentary behavior, and low adherence to the Mediterranean diet. Women consistently showed lower metabolic age compared to men across all variables. Among the studied factors, age had the most significant impact on metabolic age, reflecting the natural decline in biological maintenance and repair systems as part of the aging process [38].

Sex also plays a crucial role in determining metabolic age. Men generally have greater muscle mass and lower fat mass compared to women, which could initially be advantageous in terms of basal metabolism. However, men are also more predisposed to accumulating visceral fat, which is associated with a higher risk of metabolic and cardiovascular diseases [39]. Women, especially after menopause, undergo hormonal changes, such as reduced estrogen levels, which can lead to increased abdominal fat and decreased muscle mass, contributing to a higher metabolic age [40]. In our study, women exhibited a lower risk of high ALLY metabolic age compared to men across all age groups. In our study, we did not account for the phase of the menstrual cycle, during which Kanellakis et al. [41] observed changes in body composition and approximately 0.5 kg of weight variation, primarily due to extracellular fluid retention during menstruation. We acknowledged this limitation, as more than half of our sample consisted of women of reproductive age.

In our study, socioeconomic level emerged as another significant determinant of metabolic health. Our findings indicate that ALLY (Avoidable Lost Life Years) increases as socioeconomic class decreases. Specifically, in social class I, values across all age groups are negative, indicating a metabolic age lower than chronological age. In contrast, in social class III, values are less negative, and in three age groups, they even become positive, with the largest differences observed in the 60–69 age group for both men and women. We observed similar trends in the prevalence of high metabolic age values, with percentages increasing alongside the individuals’ age. Multinomial logistic regression reveals an elevated risk of high ALLY metabolic age according to socioeconomic class. Therefore, in our results, socioeconomic class also shows a strong association with metabolic age. Although we did not find other studies establishing this association, we believe it could be influenced by the fact that individuals in lower socioeconomic classes often have limited access to health resources, education, and healthy food, potentially impacting their metabolic age negatively [42]. Studies have shown that stress associated with disadvantaged socioeconomic conditions can lead to unhealthy behaviors [43], such as poor diet and lack of physical activity, which increase the risk of metabolic diseases. Additionally, limited access to preventive health services and adequate treatments can exacerbate these conditions [44].

Smoking is a well-known risk factor for numerous chronic diseases, including metabolic and cardiovascular conditions [45]. Nicotine and other toxic substances in tobacco negatively affect the metabolism by increasing insulin resistance, causing systemic inflammation, and altering lipid profiles [46]. Smoking is also associated with reduced muscle mass and increased visceral fat, factors that contribute to a higher metabolic age [47]. Various studies have shown that smokers have a higher risk of developing type 2 diabetes and metabolic syndrome, which accelerates their metabolic aging [48]. Our study found that smoking significantly impacted metabolic age, with a substantial ALLY index that was more pronounced in men but statistically significant for both sexes. High metabolic age was more prevalent among smokers of both genders, with metabolic age progressively increasing with age in the smoking group. This trend was consistently higher in men across all age groups. Multinomial logistic regression analysis confirmed that smoking significantly increased the risk of high metabolic age across all age groups.

Regular physical activity has been shown to reduce the risk of at least 26 chronic diseases, promoting health and quality of life. Similarly, a lack of physical activity has been linked to an increased risk of disease. Studies demonstrate that the more regular physical activity or exercise one engages in, the greater the health benefits. Moreover, regular physical activity can induce metabolic changes that lead to improved cardiometabolic health [49]. Thus, regular physical activity is one of the most effective factors in reducing metabolic age. Exercise improves insulin sensitivity, reduces visceral fat, and increases muscle mass, all of which are beneficial to metabolic health [50]. Additionally, regular physical activity reduces systemic inflammation and improves lipid profiles, which can counteract the negative effects of chronological aging [51]. In our study, physical activity was the variable most strongly associated with ALLY. Both in terms of average metabolic age values and the prevalence of high metabolic age, it was the factor that showed the greatest differences between those who did not engage in regular physical activity and those who maintained a high level of exercise. These results were consistent across all age groups and in both men and women. The largest differences were observed in the older age groups for both sexes, which may be related to the importance of lifelong physical exercise in mitigating the effects of biological aging.

Although no single diet is universally beneficial for health, a healthy diet emphasizes following dietary guidelines rather than restricting specific nutrients [52]. In our study, we assessed the Mediterranean diet. This diet, rich in fruits, vegetables, whole grains, olive oil, fish, and nuts, has been associated with numerous benefits for metabolic health. It is characterized by high levels of antioxidants, fiber, and healthy fats, which help reduce inflammation and improve insulin sensitivity [53].

The Mediterranean diet has demonstrated benefits for cardiovascular disease, diabetes mellitus, metabolic syndrome, overweight and obesity, various cancers, cognitive function, and overall mortality reduction. Further, the Mediterranean diet is linked to improved body composition and reduced visceral fat accumulation, key factors for achieving optimal metabolic health [54]. However, to our knowledge, this is the first study linking the Mediterranean diet with metabolic age. Our results show that adherence to a Mediterranean diet has a substantial influence on metabolic age, with the highest OR observed across all age ranges following physical activity.

In the results of our study, we observed that ALLY, in both men and women, showed a significant difference between those who adhered to the Mediterranean diet and those who did not. However, in this case, the differences were greater in men across all age groups compared to women. In the multinomial logistic regression, we found that lack of adherence to the Mediterranean diet increased the risk of high ALLY metabolic age from 184% in the 30–39 age group to 444% in the oldest age group. This makes the Mediterranean diet the second most influential variable in determining an individual’s metabolic age.

Given the importance of these findings, it is essential to promote adherence to the Mediterranean diet across all age groups, which requires appropriate health and nutrition promotion policies and interventions. This is particularly urgent, as a systematic review conducted by Obeid et al. in 2022 reported a significant decline in adherence to the Mediterranean diet in Mediterranean Basin countries [55].

Similarly, a study conducted in Spain by Herrera-Ramos et al. in 2023 also identified a decline in adherence to the Mediterranean diet among Spanish children and adolescents over the past 20 years. Although the adherence levels reported in this study were slightly higher than those observed in our study, the decreasing consumption of key Mediterranean foods is concerning. For instance, the frequency of fish consumption two to three times per week dropped from approximately 80% in the 1998–2000 cohort to 65% in the 2019–2020 cohort. Likewise, legume consumption decreased from 90% in the 1998–2000 cohort to about 55% in the 2019–2020 cohort [56].

This study highlights the deterioration of dietary habits among Spanish children and adolescents and underscores the urgent need to ensure a high-quality diet. It is crucial for children and adolescents to adopt a Mediterranean diet, as it benefits their health and quality of life. Furthermore, developing healthy dietary habits during childhood increases the likelihood of their continuation into adulthood.

The global population aged over 65 is increasing, and it is projected to reach 2.1 billion people by 2050 [57]. Aging is associated with increased morbidity and changes in body composition, including an increase in fat mass, a decrease in muscle mass, and a reduction in basal metabolic rate (BMR) [58]. This situation is linked to obesity and numerous related pathologies [59], leading to a significant decline in quality of life and rising costs for society and the healthcare system, which may become unsustainable. Consequently, it is crucial to seek effective strategies that promote healthy aging and help mitigate morbidity, thereby enhancing quality of life during these additional years.

Among these strategies is the potential reduction in metabolic age by targeting various associated factors, which we have evaluated in this study. In this context, it is important to emphasize that a metabolic age lower than chronological age increases BMR, thereby reducing mortality and morbidity [60]. In our study, we examined modifiable variables associated with metabolic age that could be targeted to improve it. These variables were assessed across different age and sex strata to understand how each factor influences these groups and to determine if focusing on specific factors by sex and age group would result in more efficient health interventions.

Public spending increases with age due to the growing health and social care needs of the elderly, which are influenced by their diminished ability to manage daily living activities and reduced physical capacities [61]. Therefore, resource needs for older individuals are more closely related to disability than chronological age. One of the primary determinants of healthcare costs is the health status of the population. Addressing modifiable risk factors associated with non-communicable diseases is a cost-effective strategy to reduce their burden. These risk factors include harmful alcohol consumption, smoking, unhealthy diets, and physical inactivity—all of which also contribute to metabolic age. Every dollar invested in interventions aimed at reducing these risk factors could potentially generate a savings of 7 dollars (USD) [62].

In our study, physical inactivity emerged as the most influential variable affecting metabolic age. A study published in The Lancet in 2016 estimated the global health-related costs of physical inactivity at 53.8 billion international dollars (INT), a value representing the amount of goods and services that an individual or government could purchase in their respective country compared to what could be acquired in the United States [63,64]. Another study published in The Lancet in 2023 estimated the global cost of physical inactivity to be approximately 520 billion INT$ over the 2020–2030 decade [65].

Preventing disease earlier in life leads to a better quality of life and lower healthcare costs. Interventions aimed at reducing needs in old age should be implemented across all age groups, as they can benefit the healthcare system. The dissemination and implementation of strategies to improve health are essential to reduce social and healthcare costs. In an era of increasing budgetary constraints, analyzing costs and identifying the factors that influence them could help define strategies for cost reduction.

## 5. Strengths and Limitations

One of the strengths of the study is the large sample size, nearly 8600 people, and the wide variety of variables analyzed.

The main limitation is that only individuals of working age (18–69 years) were included, thus excluding unemployed individuals, retirees, those under 18, and those over 69 years old. Our results cannot be extrapolated to the entire population as some age groups are missing.

Since more than half of the sample belongs to social class I, it may not be representative of the general population

Likewise, since this is a population of the Balearic Islands only, it is possible that the results may differ in other types of populations. Therefore, our results cannot be extrapolated to them.

Other confounding factors such as comorbidities or pharmacological treatments were not included, as these data were not available.

Another limitation of our study is the healthy worker bias, which is a common methodological problem in research studies on workers. Workers with chronic diseases or who are more prone to illness may be less likely to attend occupational health check-ups compared to healthy workers, which could underestimate the results.

Since this is a cross-sectional study, it allows for an association between the variables studied and obesity, but we cannot establish causality.

Finally, we have not taken into account the menstrual phase of the women included in the study, which may produce variations in their weight.

## 6. Conclusions

Metabolic age is shaped by both demographic and lifestyle factors. While chronological age and sex are unmodifiable influences, modifiable factors like physical activity and adherence to the Mediterranean diet have the most significant impact, followed by socioeconomic status and smoking habits. Addressing these modifiable factors through targeted interventions and public health policies is crucial to improving metabolic health, reducing chronic disease burdens, lowering healthcare costs, and enhancing overall quality of life.

## Figures and Tables

**Figure 1 nutrients-16-04207-f001:**
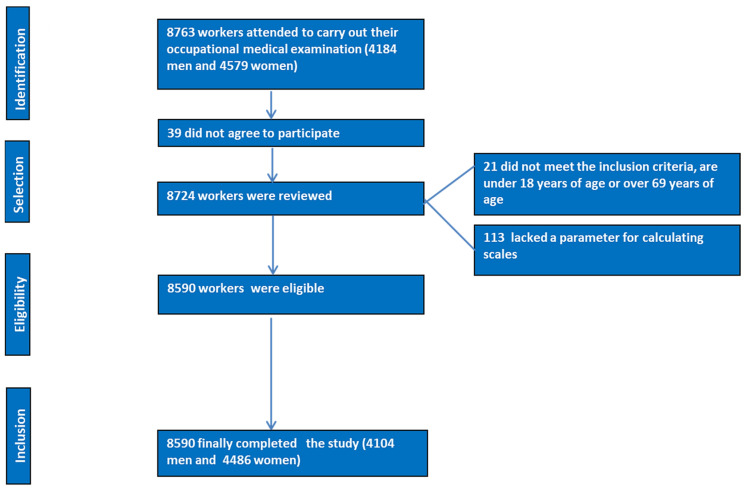
PRISMA flowchart of participants in this study.

**Table 1 nutrients-16-04207-t001:** Characteristics of the participants.

	Men n = 4104	Women n = 4486	
	Mean (SD)	Mean (SD)	*p*-Value
**Age (years)**	41.6 (10.6)	41.5 (10.5)	0.492
**Height (cm)**	175.8 (7.2)	162.5 (6.1)	<0.001
**Weight (kg)**	81.2 (14.8)	63.9 (13.6)	<0.001
**Waist circumference (cm)**	89.8 (12.5)	77.0 (12.0)	<0.001
**Hip circumference (cm)**	101.8 (8.7)	99.6 (10.9)	<0.001
**Systolic blood pressure (mmHg)**	128.6 (13.3)	117.2 (14.1)	<0.001
**Diastolic blood pressure (mmHg)**	79.9 (10.2)	74.9 (9.9)	<0.001
**Glycaemia (mg/dL)**	93.4 (17.8)	88.9 (12.6)	<0.001
**Total cholesterol (mg/dL)**	191.8 (36.0)	189.0 (34.8)	<0.001
**HDL-cholesterol (mg/dL)**	49.2 (11.3)	59.5 (12.8)	<0.001
**LDL-cholesterol (mg/dL)**	124.0 (54.6)	113.8 (30.7)	<0.001
**Triglycerides (mg/dL)**	107.8 (69.4)	81.5 (46.3)	<0.001
**GGT (UI)**	31.5 (30.0)	18.5 (15.9)	<0.001
**AST (UI)**	24.4 (17.3)	18.2 (7.7)	<0.001
**ALT (UI)**	29.3 (34.9)	17.3 (13.4)	<0.001
	**%**	**%**	***p*-value**
**18–29 years**	15.5	16.8	0.005
**30–39 years**	27.8	25.1	
**40–49 years**	32.7	34.4	
**50–59 years**	19.0	19.7	
**60–69 years**	5.0	4.0	
**Social class I**	57.1	50.8	<0.001
**Social class II**	20.2	23.8	
**Social class III**	22.7	25.4	
**Non-smokers**	84.5	84.2	0.348
**Smokers**	15.5	15.8	
**Physical inactivity**	25.9	35.1	<0.001
**PhA Moderate**	27.0	26.5	
**PhA High**	47.1	38.4	
**NAD Mediterranean diet**	44.5	41.6	<0.001
**AD Mediterranean diet**	55.5	58.4	

SD Standard deviation. HDL-c High-density lipoprotein. LDL-c Low-density lipoprotein. GGT Gamma-Glutamyl Transferase. AST Aspartate Aminotransferase. ALT Alanine Aminotransferase. PhA Physical activity. AD Mediterranean diet: Adherence to the Mediterranean diet. NAD Mediterranean diet: No Adherence to the Mediterranean diet. Student’s *t*-test was used for means and chi square test for prevalence.

**Table 2 nutrients-16-04207-t002:** Mean metabolic age values stratified by age and according to different sociodemographic variables and healthy habits by sex.

		18–29 Years		30–39 Years		40–49 Years		50–59 Years		60–69 Years
Men	n	Mean (SD)	n	Mean (SD)	n	Mean (SD)	n	Mean (SD)	n	Mean (SD)
**Social class I**	474	−5.2 (10.5) *	774	−6.6 (9.6) *	630	−5.5 (10.6) *	324	−3.8 (10.8) *	144	−3.9 (11.1) *
**Social class II**	84	−3.7 (6.7) *	204	−0.1 (10.9) *	300	−3.5 (11.3) *	216	−3.4 (9.9) *	24	2.0 (11.0) *
**Social class III**	78	−2.5 (10.4) *	162	1.4 (13.3) *	414	−2.4 (11.2) *	240	0.7 (12.5) *	36	8.0 (7.2) *
**Non-smokers**	534	−5.7 (9.6) *	924	−4.4 (11.1) *	1158	−4.6 (10.9) *	672	−2.5 (11.1) *	180	−1.5 (11.5) *
**Smokers**	102	0.5 (11.2) *	216	0.2 (10.6) *	186	−0.7 (11.3) *	108	1.3 (12.2) *	24	1.3 (11.1) *
**Physical inactivity**	120	1.7 (11.2) *	270	1.4 (11.9) *	390	3.7 (10.8) *	198	3.8 (9.9) *	84	6.0 (9.6) *
**PhA Moderate**	144	−6.3 (8.8) *	348	−2.8 (10.7) *	294	−4.6 (8.5) *	264	−0.7 (11.2) *	60	−5.5 (8.7) *
**PhA High**	372	−8.2 (9.5) *	522	−8.3 (8.9) *	660	−8.5 (9.6) *	318	−7.4 (9.8) *	60	−7.9 (10.2) *
**NAD Mediterranean diet**	225	−1.4 (11.3) *	426	−1.1 (12.1) *	610	−0.2 (12.3) *	433	1.4 (11.4) *	133	3.4 (10.6) *
**AD Mediterranean diet**	411	−6.4 (8.9) *	714	−6.2 (9.8) *	734	−7.3 (8.6) *	347	−6.9 (9.3) *	71	−10.6 (6.1) *
**Women**	**n**	**Mean (SD)**	**n**	**Mean (SD)**	**n**	**Mean (SD)**	**n**	**Mean (SD)**	**n**	**Mean (SD)**
**Social class I**	606	−6.2 (9.2) *	702	−8.0 (9.7) *	682	−8.0 (9.5) *	222	−7.9 (10.2) *	66	−7.0 (10.5) *
**Social class II**	80	−3.6 (10.5) *	192	−3.9 (10.5) *	464	−2.7 (12.3) *	304	−1.9 (11.9) *	28	−1.2 (12.7) *
**Social class III**	68	−2.1 (10.8) *	232	−1.7 (11.9) *	398	−1.5 (12.2) *	356	0.6 (11.5) *	86	0.9 (11.3) *
**Non-smokers**	694	−5.3 (9.8) *	942	−6.2 (10.5) *	1304	−5.0 (11.3) *	682	−6.5 (10.2) *	154	−5.0 (11.9) *
**Smokers**	60	−4.3 (10.5) *	184	−3.6 (11.3) *	240	−3.3 (12.5) *	200	−2.0 (10.4) *	26	0.2 (10.7) *
**Physical inactivity**	192	−3.1 (11.3) *	366	−1.1 (11.5) *	600	0.3 (12.1) *	356	0.5 (12.2) *	60	0.6 (12.1) *
**PhA Moderate**	238	−5.5 (8.8) *	330	−6.9 (9.7) *	376	−5.6 (10.5) *	209	−5.1 (10.9) *	34	−7.2 (11.0) *
**PhA High**	324	−6.2 (9.5) *	430	−9.5 (9.0) *	568	−9.5 (9.0) *	317	−8.6 (9.5) *	86	−10.4 (8.3) *
**NAD Mediterranean diet**	297	−3.2 (11.0) *	425	−4.9 (11.3) *	653	−2.7 (12.1) *	399	−2.5 (12.4) *	92	−0.9 (12.0) *
**AD Mediterranean diet**	457	−6.5 (8.8) *	701	−6.6 (10.2) *	891	−6.2 (10.7) *	483	−6.7 (10.2) *	88	−8.7 (10.0) *

PhA Physical activity. AD Mediterranean diet: Adherence to the Mediterranean diet. NAD Mediterranean diet: No Adherence to the Mediterranean diet. SD Standard deviation. (*) *p* < 0.001. Student’s *t*-test was used.

**Table 3 nutrients-16-04207-t003:** Prevalence of high metabolic age values stratified by age and according to different sociodemographic variables and healthy habits by sex.

		18–29 Years		30–39 Years		40–49 Years		50–59 Years		60–69 Years
Men	n	%	n	%	n	%	n	%	n	%
**Social class I**	474	13.2 *	774	15.5 *	630	20.8 *	324	22.2 *	144	24.8 *
**Social class II**	84	21.5 *	204	25.3 *	300	28.0 *	216	30.9 *	24	34.5 *
**Social class III**	78	23.1 *	162	28.1 *	414	34.8 *	240	35.4 *	36	39.9 *
**Non-smokers**	534	20.7 *	924	23.5 *	1158	25.9 *	672	27.7 *	180	30.9 *
**Smokers**	102	28.9 *	216	28.8 *	186	32.7 *	108	33.3 *	24	38.9 *
**Physical inactivity**	120	40.0 *	270	44.4 *	390	46.9 *	198	50.3 *	84	54.6 *
**PhA Moderate**	144	14.5 *	348	17.6 *	294	18.4 *	264	21.8 *	60	26.8 *
**PhA High**	372	12.2 *	522	12.6 *	660	14.5 *	318	15.8 *	60	18.2 *
**NAD Mediterranean diet**	225	32.0 *	426	37.1 *	610	40.8 *	433	42.3 *	133	44.9 *
**AD Mediterranean diet**	411	15.7 *	714	19.4 *	734	21.5 *	347	24.2 *	71	27.9 *
**Women**	**n**	**%**	**n**	**%**	**n**	**%**	**n**	**%**	**n**	**%**
**Social class I**	606	10.5 *	702	12.2 *	682	13.8 *	222	16.2 *	66	18.2 *
**Social class II**	80	19.2 *	192	20.8 *	464	24.7 *	304	27.6 *	28	28.6 *
**Social class III**	68	20.6 *	232	26.8 *	398	28.9 *	356	29.2 *	86	31.8 *
**Non-smokers**	694	19.3 *	942	18.9 *	1304	20.5 *	682	23.3 *	154	23.4 *
**Smokers**	60	23.3 *	184	23.9 *	240	27.8 *	200	29.9 *	26	32.3 *
**Physical inactivity**	192	29.2 *	366	34.4 *	600	38.8 *	356	43.0 *	60	43.3 *
**PhA Moderate**	238	12.8 *	330	14.9 *	376	19.1 *	209	21.1 *	34	23.5 *
**PhA High**	324	8.7 *	430	10.7 *	568	11.6 *	317	13.2 *	86	14.6 *
**NAD Mediterranean diet**	297	20.3 *	425	22.6 *	653	30.5 *	399	33.3 *	92	38.3 *
**AD Mediterranean diet**	457	14.6 *	701	18.0 *	891	19.9 *	483	21.8 *	88	23.2 *

PhA Physical activity. AD Mediterranean diet: Adherence to the Mediterranean diet. NAD Mediterranean diet: No Adherence to the Mediterranean diet. (*) *p* < 0.001. The chi-square test was used.

**Table 4 nutrients-16-04207-t004:** Multinomial logistic regression stratified by age.

	18–29 Years	30–39 Years	40–49 Years	50–59 Years	60–69 Years
ALLY Metabolic Age High	OR (95% CI)	OR (95% CI)	OR (95% CI)	OR (95% CI)	OR (95% CI)
**Women**	1	1	1	1	1
**Men**	1.33 (1.28–1.39) *	1.74 (1.50–1.98) *	1.36 (1.29–1.44) *	1.29 (1.23–1.35) *	1.24 (1.20–1.29) *
**Social class I**	1	1	1	1	1
**Social class II**	1.19 (1.16–1.23) *	1.59 (1.45–1.74) *	1.71 (1.54–1.89) *	2.47 (1.99–2.96) *	1.45 (1.31–1.59) *
**Social class III**	1.54 (1.40–1.68) *	3.54 (2.71–4.34) *	2.58 (2.06–3.11) *	2.74 (2.03–3.45) *	1.56 (1.38–1.75) *
**Non-smokers**	1	1	1	1	1
**Smokers**	2.13 (1.82–2.45) *	1.45 (1.38–1.53) *	1.14 (1.10–1.17) *	1.31 (1.26–1.37) *	1.23 (1.19–1.27) *
**PhA High**	1	1	1	1	1
**PhA Moderate**	2.69 (2.28–3.09) *	2.12 (1.78–2.46) *	3.39 (2.65–4.14) *	1.61 (1.39–1.83) *	3.86 (2.99–4.74) *
**Physical inactivity**	2.99 (2.14–3.85) *	4.60 (3.51–5.70) *	7.11 (5.67–8.55) *	4.54 (3.36–5.73) *	8.36 (6.90–9.82) *
**AD Mediterranean diet**	1	1	1	1	1
**NAD Mediterranean diet**	2.84 (2.13–3.55) *	1.84 (1.48–2.20) *	3.33 (2.75–3.89) *	3.39 (2.63–4.15) *	4.44 (3.05–5.84) *

PhA Physical activity. AD Mediterranean diet: Adherence to the Mediterranean diet. NAD Mediterranean diet: No Adherence to the Mediterranean diet. OR: Odds ratio. CI: Confidence interval. Statistical differences (*) *p* < 0.001.

## Data Availability

This study data are stored in a database that complies with all security measures at ADEMA-Escuela Universitaria. The Data Protection Delegate is Ángel Arturo López González.

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
