# Peer review of "Associations Between Metabolic Age, Sociodemographic Variables, and Lifestyle Factors in Spanish Workers"

_nutrients, 2024, doi:10.3390/nu16234207_

Round 1

Reviewer 1 Report (Previous Reviewer 1)

Comments and Suggestions for Authors

The study Associations Between Metabolic Age, Sociodemographic Variables, and Lifestyle Factors in Spanish Workers represents a response to the questions raised.

* In this sense, when observing the considerations presented by the authors, I emphasize that the study is good quality and robust. Despite delivering a fundamental methodology, it is still relevant.

* As the authors work with anthropometric variables and body composition, reflecting on women's menstrual periods during the assessments would be essential. This fact directly interferes with the values ​​and can lead to a false positive. It is necessary to present a discussion on this point.

Author Response

The study Associations Between Metabolic Age, Sociodemographic Variables, and Lifestyle Factors in Spanish Workers represents a response to the questions raised.

* In this sense, when observing the considerations presented by the authors, I emphasize that the study is good quality and robust. Despite delivering a fundamental methodology, it is still relevant.

First, we would like to thank the reviewer for recognizing the effort made by the authors. Secondly, we appreciate your new suggestions to improve the article.

Below, we respond to each of your recommendations.

* As the authors work with anthropometric variables and body composition, reflecting on women's menstrual periods during the assessments would be essential. This fact directly interferes with the values ​​and can lead to a false positive. It is necessary to present a discussion on this point.

You are absolutely correct. We have included a paragraph with relevant references in the discussion section. Additionally, we have addressed this point in the study's limitations. Thank you very much for your valuable observation.

Dear reviewer, we have proceeded to answer all the questions raised and have made changes to the manuscript. To facilitate their location in the paper, we have highlighted the modifications in red.

We trust that the answers will be to your liking and we appreciate all the recommendations made to improve our manuscript.

Reviewer 2 Report (New Reviewer)

Comments and Suggestions for Authors

This paper presents a study that aimed at linking metabolic age (as a biological variable) with sociodemographic and lifestyle characteristics. The study was conducted on a significant sample, however from a quite limited area and age group (workers on Balearic islands).

Presented data are not a significant novelty, given that we already know that diet and physical activity influence health. Also, statistical methods employed are very descriptive and basic. However, the introduction of an objective, biological variable such as metabolic age is an added value of the study.

It is surprising that the study does not mention other studies on Mediterranean diet adherence conducted in Spain, which show a relatively negative trend in dietary patterns (Herrera-Ramos et al. 2023; Obeid et al. 2022). Previous studies show that, over the past decade, adherence to the Mediterranean diet in Spain exhibits a concerning decline and this is not supported by results of this study, which state that more than half of the sample (both genders) adhere to the Meditteranean diet. This is not a problem per se, but it would be important to address this difference from a trend detected in other Spanish studies.

Also, given the size of the cohort, it would have been beneficial to have used a machine learning approach. Machine learning algorithms can analyze multiple variables simultaneously, potentially uncovering complex interactions between SES, lifestyle factors, and metabolic age. Also, ML methods can identify nonlinear patterns that traditional statistical methods might miss.

  •  

Author Response

This paper presents a study that aimed at linking metabolic age (as a biological variable) with sociodemographic and lifestyle characteristics. The study was conducted on a significant sample, however from a quite limited area and age group (workers on Balearic islands).

Presented data are not a significant novelty, given that we already know that diet and physical activity influence health. Also, statistical methods employed are very descriptive and basic. However, the introduction of an objective, biological variable such as metabolic age is an added value of the study.

First, we would like to thank the reviewer for recognizing the effort made by the authors. Secondly, we appreciate your new suggestions to improve the article.

Below, we respond to each of your recommendations.

It is surprising that the study does not mention other studies on Mediterranean diet adherence conducted in Spain, which show a relatively negative trend in dietary patterns (Herrera-Ramos et al. 2023; Obeid et al. 2022). Previous studies show that, over the past decade, adherence to the Mediterranean diet in Spain exhibits a concerning decline and this is not supported by results of this study, which state that more than half of the sample (both genders) adhere to the Meditteranean diet. This is not a problem per se, but it would be important to address this difference from a trend detected in other Spanish studies.

Thank you very much for your insightful reflection. Following your recommendations, we have added paragraphs in the discussion section addressing the negative trend in adherence to the Mediterranean diet. When comparing our findings with those of the study by Herrera-Ramos et al., we observed no significant differences, as we reported adherence rates slightly above 50% for both sexes. Similarly, Herrera-Ramos et al. highlighted a substantial decline in adherence to the Mediterranean diet between the two cohorts studied, reporting fish consumption 2–3 times per week at 65.4% for men and 66.1% for women in the second cohort. Likewise, legume consumption was reported at 68.7% for men and 69.6% for women in the 2019–2020 cohort. These findings have been duly incorporated into our discussion section.

Once again, we appreciate your comment, as we believe it is crucial to reflect on the significance of the Mediterranean diet and the serious problem posed by the decreasing adherence to it.

Also, given the size of the cohort, it would have been beneficial to have used a machine learning approach. Machine learning algorithms can analyze multiple variables simultaneously, potentially uncovering complex interactions between SES, lifestyle factors, and metabolic age. Also, ML methods can identify nonlinear patterns that traditional statistical methods might miss.

You are absolutely right, and we appreciate your insight. The use of artificial intelligence can provide significant benefits to research. We will keep this in mind for future studies. Thank you very much.

Dear reviewer, we have proceeded to answer all the questions raised and have made changes to the manuscript. To facilitate their location in the paper, we have highlighted the modifications in red.

We trust that the answers will be to your liking and we appreciate all the recommendations made to improve our manuscript.

This manuscript is a resubmission of an earlier submission. The following is a list of the peer review reports and author responses from that submission.

Round 1

Reviewer 1 Report

Comments and Suggestions for Authors

Dear Authors

The study has a good sample. However, the methods are fragile.

The authors attempt to establish a metabolic age relationship, but do not present any parameters for this relationship. Only a subdivision of the sample into age and sex. This comparison doesn't make much sense. This type of study must compare the different parameters at different ages to finally create a reference table. The authors just describe findings without much focus. Blood parameters are basic. They could have measured cytokines or marked specific oxidative or metabolic stress. The level of physical activity is very subjective, which does not allow for any conclusions.

Author Response

Dear Reviewer,

Thank you very much for your observations. We have modified Table 2 and Table 3 accordingly. In Table 2, we present the metabolic age values stratified by age, sex, and different variables. Negative values indicate that metabolic age is lower than biological age, showing the association with each of the studied variables. There may be a confounding factor, as some individuals in the study may present several of these variables simultaneously. In Table 3, we have described the percentage of high metabolic age values stratified by age and differentiated by sex, in an attempt to clarify this concept. Finally, in Table 4, we performed a multinomial logistic regression stratified by age to avoid the confounding factors of different variables and to determine the influence of each (increase in risk) across different age groups.

Regarding the measurement of cytokines, oxidative or metabolic stress, you are correct; it would have been interesting to perform these measurements. However, as we lack this data, we cannot include it in our study.

As for the level of physical activity, you are right; our doctoral student did not specify that the International Physical Activity Questionnaire (IPAQ), which is internationally validated, was used to assess physical activity levels. We have corrected this in the manuscript.

We appreciate your understanding and patience.

Dear reviewer, we have proceeded to answer all the questions raised and have made changes to the manuscript. To facilitate their location in the paper, we have highlighted the modifications in red.

We trust that the answers will be to your liking and we appreciate all the recommendations made to improve our manuscript.

Reviewer 2 Report

Comments and Suggestions for Authors

The manuscript is very well constructed and the research appears to have been executed professionally and sensitively.

Results are meaningful and suggest new directions for public health measurement and research.

Although the document does provide some hints of implications of the findings, the broder impact of the research would be enhanced by a couple of additional paragraphs detailing more specific implications for policy and practice.

The results are strong, with most p-values very small owing to the large sample size. The impact of the model estimation would be enhanced by sharing model-scale fit metrics beyond the odds ratios which provide metrics for individual model components.

Author Response

Estimado revisor,

Muchas gracias por sus observaciones.

Si bien el documento ofrece algunas pistas sobre las implicaciones de los hallazgos, el impacto más amplio de la investigación se vería reforzado por un par de párrafos adicionales que detallaran implicaciones más específicas para las políticas y la práctica.

Siguiendo sus recomendaciones, hemos añadido algunos párrafos sobre las implicaciones políticas, socio-sanitarias y de práctica clínica.

Muchas gracias.

Los resultados son sólidos, ya que la mayoría de los valores p son muy pequeños debido al gran tamaño de la muestra. El impacto de la estimación del modelo se vería mejorado si se compartieran métricas de ajuste a escala del modelo más allá de los odds ratios, que proporcionan métricas para los componentes individuales del modelo.

Hemos modificado las tablas y las hemos estratificado por edad y sexo. Creemos que esto mejora el modelo, tal como nos has aconsejado.

Muchas gracias por tu recomendación.

Estimado revisor, hemos procedido a responder todas las preguntas planteadas y hemos realizado cambios en el manuscrito. Para facilitar su localización en el artículo, hemos resaltado las modificaciones en rojo.

Confiamos en que las respuestas sean de su agrado y agradecemos todas las recomendaciones realizadas para mejorar nuestro manuscrito.

Reviewer 3 Report

Comments and Suggestions for Authors

Dear authors this is a very interesting study on the variables associated with the metabolic age of a large sample of working persons in the Balearic islands. The manuscript is well-written and comprehensive in most sections.

Please find a few comments and suggestions:

- The study is a cross-sectional study (as mentioned in the Methodology section). Due the study nature, I suggest that the article title should rather refer to associations between … and not to “variables that affect”, which implies a causality statement. This should also be added to the study limitations.

- In my opinion, in Table 1, except for the chronological age, you should provide the respective values for the metabolic age of the participants. In addition, the percentage of individuals with high metabolic age should be added to the Table for the whole group and the subgroups.
- I was confused by reading Table 2 in your manuscript. You mention that Table 2 presents the mean values of metabolic age stratified by each of the sociodemographic variables and health-related habits. However, the values for the metabolic age presented are negative. In my opinion, the title of this Table does not describe clearly the values presented. 
- I faced the same problem while reading the discussion sessions where you were interpreting these values. While your conclusions were understandable, the negative values of the metabolic age in Table 2 created a difficulty in following your interpretation of those results. Furthermore, in the Discussion section you refer to the years of potential life lostOur results show an increase in years of potential life lost (YPLL) as social class decreased, with men experiencing a rise from an average of -5.5 years in social class I to -0.8 years in social class III”. I assume that with “years of potential life lost” you mean the difference between chronological and metabolic age. This should be better explained. Again, in my opinion, the negative values are confusing. Perhaps, it would be better to mention the actual mean values of the metabolic age in each subgroup of the participants.

Kind regards

Author Response

Dear reviewer, we appreciate your recommendation and have proceeded to modify the title according to your suggestions. We hope it meets your approval: 'Metabolic Age and Its Association with Biological Age, Sex, Social Class, and Lifestyle Habits.'

You are correct, and following your recommendation, we have added this to the study's limitations: 'Since this is a cross-sectional study, it allows for an association between the variables studied and obesity, but we cannot establish causality.'

- In my opinion, in Table 1, except for the chronological age, you should provide the respective values for the metabolic age of the participants. In addition, the percentage of individuals with high metabolic age should be added to the Table for the whole group and the subgroups.

Following your advice, we have attempted to clarify both the tables and the results. We have proceeded to modify Tables 2, 3, and 4.

We did not make any changes to Table 1 in order to avoid conflicting with the clarifications requested by you and another reviewer, who advised against repeating data across different tables. Therefore, what you requested for Table 1 has been added to Table 3. We appreciate your understanding.

- I was confused by reading Table 2 in your manuscript. You mention that Table 2 presents the mean values of metabolic age stratified by each of the sociodemographic variables and health-related habits. However, the values for the metabolic age presented are negative. In my opinion, the title of this Table does not describe clearly the values presented.

We have modified Table 2 and Table 3. In Table 2, we present the metabolic age values stratified by age, according to different variables and sex. Negative values indicate that the metabolic age is lower than the biological age, showing the association with each of the studied variables. These results may contain a confounding factor, as some individuals in the study may exhibit multiple variables simultaneously. In Table 3, we describe the percentage of high metabolic age values stratified by age and differentiated by sex, in an attempt to clarify this concept. Finally, in Table 4, we performed multinomial logistic regression stratified by age, in order to eliminate confounding factors between the various variables and to observe the influence of each variable (risk increase) across different age groups.

- I faced the same problem while reading the discussion sessions where you were interpreting these values. While your conclusions were understandable, the negative values of the metabolic age in Table 2 created a difficulty in following your interpretation of those results. Furthermore, in the Discussion section you refer to the years of potential life lost “Our results show an increase in years of potential life lost (YPLL) as social class decreased, with men experiencing a rise from an average of -5.5 years in social class I to -0.8 years in social class III”. I assume that with “years of potential life lost” you mean the difference between chronological and metabolic age. This should be better explained. Again, in my opinion, the negative values are confusing. Perhaps, it would be better to mention the actual mean values of the metabolic age in each subgroup of the participants.

Thank you for your observation. We have modified Table 2 and the discussion text to make them more comprehensible. We were unable to remove the negative values, as they accurately reflect the difference between metabolic age and biological age, indicating a lower metabolic age compared to the corresponding biological age. Nevertheless, we have tried to clarify the presentation as much as possible and have revised the discussion.

Thank you very much.

Dear reviewer, we have proceeded to answer all the questions raised and have made changes to the manuscript. To facilitate their location in the paper, we have highlighted the modifications in red.

We trust that the answers will be to your liking and we appreciate all the recommendations made to improve our manuscript.

Reviewer 4 Report

Comments and Suggestions for Authors

This study examined the impact of different sociodemographic variables and lifestyle habits on metabolic age values in a cohort of workers from the Balearic Islands. Overall, it is an informative paper that could make a valuable contribution to the field. However, before further consideration, the authors should revise the manuscript to address the following issues. Furthermore, the entire manuscript should be carefully proofread to ensure proper English and scientific writing. Line numbers have been provided where relevant for reference.

Major Concern (Statistical Analysis Strategy):

The main concern relates to the statistical analysis strategy, which forms the foundation of the study’s conclusions. The methods section (Lines 167–170) provides an unclear and insufficiently detailed description of the logistic regression analysis. Although the authors mention using multinomial logistic regression to identify variables linked to significant risk factors, they do not offer enough information on how the analysis was executed. Moreover, the rationale for selecting multinomial logistic regression is not well justified. If the outcome variable was divided into more than two categories, this should be clearly stated. Additionally, there is no clear explanation of how potential confounders were identified and accounted for in the logistic regression model, which is crucial to avoid biased estimates in such analyses.

Other Important Issues:

1.  Title:

The current title is somewhat broad and lacks specificity. It should be refined to improve clarity and focus, particularly as a scientific article. For instance, the term “variables” is too general and does not specify the types of variables being examined. If the sample size is significant, it could be included, but in this reviewer's opinion, it may be more appropriate to mention this in the abstract and/or methods section instead.

2. Abstract:

a) The stated aim of the study in the abstract is too broad. It could be made more specific by identifying the particular sociodemographic variables and lifestyle habits that were examined.

b) “All the variables studied affected the risk of presenting a high metabolic age” (L30-31) is a broad claim. Not all variables are likely to have the same level of impact, and this needs to be more clearly articulated with consideration of confounding variables.

c) The abstract (or the text elsewhere) does not clarify how metabolic age was categorized or why the specific cutoff (e.g., 12 years above chronological age) was chosen. Providing this information would enhance clarity.

3. Introduction:

a) The introduction does not sufficiently discuss the gaps in the existing literature. While it mentions factors like genetic factors, physical activity, diet, and stress and sleep quality, it does not critically engage with the literature to explain why the factors studied are of particular interest. Additionally, the introduction should include a more detailed discussion of what is lacking in the current body of knowledge regarding metabolic age. These considerations should guide the formulation of the specific objectives of this study.

b) The introduction provides background information on metabolic age but does not explain why it is particularly important to study it in the context of Spanish workers.

4. Materials and Methods:

a) The descriptions in the Materials and Methods section are often too general, making it difficult to replicate the study or fully assess its validity.

b) The criteria used to select the companies participating in the study are not mentioned, which leaves important methodological gaps.

c) The authors do not explicitly state any exclusion criteria beyond temporary disability, leaving it unclear how potential confounders (e.g., workers with chronic diseases or those on medication) were addressed.

d) In large datasets, missing values are common and can introduce bias if not handled properly. The authors should specify how they dealt with missing data, if applicable.

e) As noted earlier, the statistical methods section is vague and lacks essential details, making it difficult to assess the robustness of the analysis. It is not clear from the text what is meant by the expression “Variables related to the most significant risk factors were determined using multivariate methods (L167-168).

 5.  Results:

a) The presented tables are not well-organized and contain repetitive information (e.g. numbers). To improve clarity, please ensure that each table presents distinct and relevant data without unnecessary overlap. Additionally, consistent formatting and clearer titles or footnotes (such as the statistical tests used) would enhance readability and help the reader understand the data more effectively.

b) While the tables are generally understandable, the accompanying text often merely repeats the information presented in the tables. A deeper interpretation of the results in the text is needed, rather than simply summarizing the data. Please provide further insights and context for the findings.

c) With the exception of the comparison of smoking status between men and women, all p-values in the study are reported as <0.001. Please verify and confirm the accuracy of these values.

d) The meaning of phrases such as “Men had more negative anthropometric, clinical, and analytical findings" (L176), "social class I" (L177), and "Mean values were more favorable in women" (L187-188) requires clarification. Please provide more specific explanations or contextual information for these statements.

e) Although the results address various variables, the connection between the findings and the study's original aims is not always clear. Additionally, the table in the Results section presents odds ratios and confidence intervals for factors affecting metabolic age, but the statistical reporting is insufficient. This leaves readers without a clear understanding of the magnitude and direction of the effects. More detailed interpretation and statistical context are necessary.

f) The results section does not discuss whether potential confounding factors were considered in the analysis. For instance, age could confound the relationship between physical activity and metabolic age, but this possibility is neither explored nor adjusted for in the results. This omission should be addressed.

6.  Discussion

a) In the Discussion section, the authors mention years of potential life lost (YPLL), linking it to factors such as social class and physical activity. However, this concept is neither mentioned nor reported in the Results section. This represents a significant discrepancy, as introducing new terms or findings in the discussion without them being grounded in the results can confuse readers. If YPLL is an important concept, it should be explicitly presented and explained in the Results section.

b) The discussion contains redundant information and should avoid repetition (e.g., L208-212, L227-230, L244-254). Streamlining the discussion to focus on new interpretations and insights would improve clarity and readability.

c) The lack of a critical comparison between the current findings and existing literature limits the ability to assess the novelty and significance of this study. A more in-depth comparison with previous research is needed to position the study's contributions within the broader scientific context.

d) Some statements in the discussion overgeneralize the findings without sufficient qualification. For example, the broad discussion of lifestyle factors such as diet and physical activity on metabolic age lacks consideration of potential confounders like age or gender. A more nuanced discussion that acknowledges the study's limitations would strengthen the interpretation of the findings.

e) The discussion of limitations is inadequate and needs to be more comprehensive. The limitations of the cross-sectional study design and potential biases or confounding factors should be addressed to provide a clearer context for interpreting the results.

f) The potential implications of the findings for workplace health policies and interventions could be better explored. Expanding on how these results might inform health initiatives in occupational settings would enhance the practical relevance of the study.

7.   Conclusion

The Conclusion section provides a general summary of the study's findings but lacks sufficient depth. It largely repeats the points made in the discussion without offering new insights. Additionally, the conclusion includes broad statements that may appear overambitious, especially given the study’s limitations. For instance, while the authors highlight the importance of addressing modifiable lifestyle factors (L347-348), they do not adequately account for or discuss the study’s limitations, which could undermine the strength of this conclusion.

Comments on the Quality of English Language

The entire manuscript should be carefully proofread to ensure proper English and scientific writing.

Author Response

Dear Reviewer,

Thank you very much for your observations.

Major Concern (Statistical Analysis Strategy):

The main concern relates to the statistical analysis strategy, which forms the foundation of the study’s conclusions. The methods section (Lines 167–170) provides an unclear and insufficiently detailed description of the logistic regression analysis. Although the authors mention using multinomial logistic regression to identify variables linked to significant risk factors, they do not offer enough information on how the analysis was executed. Moreover, the rationale for selecting multinomial logistic regression is not well justified. If the outcome variable was divided into more than two categories, this should be clearly stated. Additionally, there is no clear explanation of how potential confounders were identified and accounted for in the logistic regression model, which is crucial to avoid biased estimates in such analyses.

We have modified the multinomial logistic regression, stratifying it by age against the studied variables as risk factors. Likewise, we have revised the explanation in the statistical analysis section. We hope this provides sufficient clarification regarding the observations you raised.

Thank you very much.

Other Important Issues:

  1. Title:

The current title is somewhat broad and lacks specificity. It should be refined to improve clarity and focus, particularly as a scientific article. For instance, the term “variables” is too general and does not specify the types of variables being examined. If the sample size is significant, it could be included, but in this reviewer's opinion, it may be more appropriate to mention this in the abstract and/or methods section instead.

Dear reviewer, we appreciate your recommendation and have proceeded to modify the title according to your suggestions. We trust it will be to your liking: 'Metabolic Age and Its Association with Biological Age, Sex, Social Class, and Lifestyle Habits.

  1. Abstract:
  2. a) The stated aim of the study in the abstract is too broad. It could be made more specific by identifying the particular sociodemographic variables and lifestyle habits that were examined.

Following your recommendations, we have specified the objective further. The aim of this study was to evaluate how age, sex, social class, smoking habits, physical activity, and adherence to the Mediterranean diet influence metabolic age.

  1. b) “All the variables studied affected the risk of presenting a high metabolic age” (L30-31) is a broad claim. Not all variables are likely to have the same level of impact, and this needs to be more clearly articulated with consideration of confounding variables.

We have removed the text "All the variables studied affected the risk of presenting a high metabolic age," leaving the odds ratios (OR) for some of the studied variables.

  1. c) The abstract (or the text elsewhere) does not clarify how metabolic age was categorized or why the specific cutoff (e.g., 12 years above chronological age) was chosen. Providing this information would enhance clarity.

In the literature review conducted, we found very few articles on metabolic age. We established the cutoff point at 12 years above chronological age based on a publication from our group cited in the article: "Ramírez Gallegos I, Marina Arroyo M, López-González ÁA, Vicente-Herrero MT, Vallejos D, Sastre-Alzamora T, Ramírez-Manent JI. The Effect of a Program to Improve Adherence to the Mediterranean Diet on Cardiometabolic Parameters in 7034 Spanish Workers. Nutrients. 2024 Apr 7;16(7):1082. doi: 10.3390/nu16071082. PMID: 38613115; PMCID: PMC11013770." and in another earlier publication: "Elguezabal-Rodelo R., Ochoa-Précoma R., Vazquez-Marroquin G., Porchia L.M., Montes-Arana I., Torres-Rasgado E., Méndez-Fernández E., Pérez-Fuentes R., Gonzalez-Mejia M.E. Metabolic age correlates better than chronological age with waist-to-height ratio, a cardiovascular risk index. Med. Clin. 2021;157:409–417. doi: 10.1016/j.medcli.2020.07.026. (In English, In Spanish)," which is included in the previous citation (L. 161). In the Materials and Methods section, point 2.2.4. Risk Scales, we have added a clarifying statement for better explanation: "Some published studies have reported that a difference of at least 12 years between chronological age and metabolic age reduces cardiovascular risk. A metabolic age 12 years or more above the chronological age was considered high."

Thank you for your observation. 

  1. Introduction:
  2. a) The introduction does not sufficiently discuss the gaps in the existing literature. While it mentions factors like genetic factors, physical activity, diet, and stress and sleep quality, it does not critically engage with the literature to explain why the factors studied are of particular interest. Additionally, the introduction should include a more detailed discussion of what is lacking in the current body of knowledge regarding metabolic age. These considerations should guide the formulation of the specific objectives of this study.

We have added a paragraph explaining the importance of metabolic age as an indicator of various response capacities of individuals to daily situations, which serves as an indirect measure of quality of life, as well as its impact on quality of life and the costs to society.

  1. b) The introduction provides background information on metabolic age but does not explain why it is particularly important to study it in the context of Spanish workers.

Thank you for your reflection. We believe that metabolic age is a truly important concept in individual health, as current knowledge suggests that it is possible to influence and reduce it. A metabolic age lower than biological age implies better quality of life and reduced healthcare costs, both for individuals and for the state. We do not believe that studying metabolic age in the population of the Balearic Islands is more important than in other populations.

Our intention has been to identify various factors that may be associated with metabolic age in the population of the Balearic Islands for several reasons: it is the population we have access to, and identifying and confirming the association between these risk factors and metabolic age allows for quick and easy implementation in primary care consultations. This could improve the quality of life for the population and decrease healthcare costs. The large sample size enhances the robustness of the results and yields a highly significant "p" value, which may prove useful for future studies in other communities or countries.

  1. Materials and Methods:
  2. a) The descriptions in the Materials and Methods section are often too general, making it difficult to replicate the study or fully assess its validity.

We have provided detailed information regarding the sample selection, exclusion criteria, and risk scales. We believe that the anthropometric measurements, as well as the clinical and analytical determinations, are clearly outlined.

We appreciate your recommendation.

  1. b) The criteria used to select the companies participating in the study are not mentioned, which leaves important methodological gaps.

The participating companies were all those that utilize our occupational health services, with the primary occupations being hospitality, construction, commerce, healthcare, public administration, transportation, education, industry, and cleaning. This is noted in the manuscript.

  1. c) The authors do not explicitly state any exclusion criteria beyond temporary disability, leaving it unclear how potential confounders (e.g., workers with chronic diseases or those on medication) were addressed.

The only exclusion criterion was that workers did not meet any of the inclusion criteria. Following your recommendations, we have included the exclusion criteria in the study.

Exclusion criteria:

  • Age under 18 years or over 69 years.
  • Not being an employee of one of the participating companies.
  • Refusal to participate in the research study.
  • Refusal to consent to the use of data for epidemiological purposes.
  • Lack of a parameter for calculating scales.

While it is true that sick workers would not attend occupational health checks, we believe that this is more of a limitation of the study rather than an exclusion criterion, and as such, we have included it in this section.

“Healthy worker bias is a common methodological problem in research studies on workers; workers with chronic illnesses or those who are more prone to illness may be less likely to attend occupational health checks compared to healthy workers, which could underestimate the results.”

We trust that you find this appropriate.

  1. d) In large datasets, missing values are common and can introduce bias if not handled properly. The authors should specify how they dealt with missing data, if applicable.

Thank you very much for your observation. As shown in the PRISMA flowchart, 113 individuals were excluded from the study due to missing data required to calculate the necessary parameters. We have included this in the exclusion criteria. Thank you for your alert.

  1. e) As noted earlier, the statistical methods section is vague and lacks essential details, making it difficult to assess the robustness of the analysis. It is not clear from the text what is meant by the expression “Variables related to the most significant risk factors were determined using multivariate methods (L167-168).

We have modified the phrase you highlighted, and we hope that this makes it clearer.

“We have carried out a multivariate analysis stratified by age groups, to evaluate the influence of each of the variables studied in each stratum”.

  1. Results:
  2. a) The presented tables are not well-organized and contain repetitive information (e.g. numbers). To improve clarity, please ensure that each table presents distinct and relevant data without unnecessary overlap. Additionally, consistent formatting and clearer titles or footnotes (such as the statistical tests used) would enhance readability and help the reader understand the data more effectively.

Following your suggestions, we have modified the tables and removed repetitive information. Additionally, we have included footnotes with the statistical tests used.

We hope this clarifies the information, and we appreciate your suggestion.

  1. b) While the tables are generally understandable, the accompanying text often merely repeats the information presented in the tables. A deeper interpretation of the results in the text is needed, rather than simply summarizing the data. Please provide further insights and context for the findings.

According to your previous recommendation, we have modified the tables and have attempted to present the results obtained in more depth in the text.

We hope this meets your approval.

  1. c) With the exception of the comparison of smoking status between men and women, all p-values in the study are reported as <0.001. Please verify and confirm the accuracy of these values.

Following your advice, we have reviewed all the statistical tests of the study. This includes all comparisons between men and women, mean metabolic age values stratified by age, prevalence of high metabolic age values stratified by age, and multinomial logistic regression stratified by age. Throughout the review conducted, the results are as presented in the paper. SPSS 29.0 software was used for the statistical analysis, as detailed in the statistical analysis section.

Thank you for your observation.

  1. d) The meaning of phrases such as “Men had more negative anthropometric, clinical, and analytical findings" (L176), "social class I" (L177), and "Mean values were more favorable in women" (L187-188) requires clarification. Please provide more specific explanations or contextual information for these statements.

We have modified the explanations of the previous sentences and have attempted to express them in a more specific and comprehensible manner.

  1. e) Although the results address various variables, the connection between the findings and the study's original aims is not always clear. Additionally, the table in the Results section presents odds ratios and confidence intervals for factors affecting metabolic age, but the statistical reporting is insufficient. This leaves readers without a clear understanding of the magnitude and direction of the effects. More detailed interpretation and statistical context are necessary.

We have modified the tables and have attempted to present in the text more clearly the impact of each of these variables on metabolic age, detailing those variables that have the greatest influence, as indicated by a higher odds ratio, thereby increasing the risk.

  1. f) The results section does not discuss whether potential confounding factors were considered in the analysis. For instance, age could confound the relationship between physical activity and metabolic age, but this possibility is neither explored nor adjusted for in the results. This omission should be addressed.

We have addressed age as a confounding factor and have modified all the tables by stratifying them by age, including age stratification in the multinomial logistic regression.

  1. Discussion
  2. a) In the Discussion section, the authors mention years of potential life lost (YPLL), linking it to factors such as social class and physical activity. However, this concept is neither mentioned nor reported in the Results section. This represents a significant discrepancy, as introducing new terms or findings in the discussion without them being grounded in the results can confuse readers. If YPLL is an important concept, it should be explicitly presented and explained in the Results section.

Thank you very much. As you recommended, we have included the ALLY (Avoidable Lost Life Years) in the results section and have discussed their significance in that section.

  1. b) The discussion contains redundant information and should avoid repetition (e.g., L208-212, L227-230, L244-254). Streamlining the discussion to focus on new interpretations and insights would improve clarity and readability.

The lines in the discussion that you suggested have been removed from the paper.

  1. c) The lack of a critical comparison between the current findings and existing literature limits the ability to assess the novelty and significance of this study. A more in-depth comparison with previous research is needed to position the study's contributions within the broader scientific context.

In our literature review, we found very few publications on metabolic age, so we based our discussion on various articles that may indirectly refer to it.

We appreciate your understanding.

  1. d) Some statements in the discussion overgeneralize the findings without sufficient qualification. For example, the broad discussion of lifestyle factors such as diet and physical activity on metabolic age lacks consideration of potential confounders like age or gender. A more nuanced discussion that acknowledges the study's limitations would strengthen the interpretation of the findings.

Thank you very much. Following your advice, we have created new tables considering age and sex. In the multinomial logistic regression stratified by age, we can see the association between the different age groups and the studied variables.

  1. e) The discussion of limitations is inadequate and needs to be more comprehensive. The limitations of the cross-sectional study design and potential biases or confounding factors should be addressed to provide a clearer context for interpreting the results.

The limitations have been reviewed, new ones have been added, and the potential confounding factors of the cross-sectional study have been detailed.

  1. f) The potential implications of the findings for workplace health policies and interventions could be better explored. Expanding on how these results might inform health initiatives in occupational settings would enhance the practical relevance of the study.

Possible improvements have been outlined, both in clinical interventions and their implications at the political and socio-health level, including the potential cost reductions that could result.

  1.  Conclusion

The Conclusion section provides a general summary of the study's findings but lacks sufficient depth. It largely repeats the points made in the discussion without offering new insights. Additionally, the conclusion includes broad statements that may appear overambitious, especially given the study’s limitations. For instance, while the authors highlight the importance of addressing modifiable lifestyle factors (L347-348), they do not adequately account for or discuss the study’s limitations, which could undermine the strength of this conclusion.

We have expanded the limitations of the study. We have also added paragraphs regarding the political, socio-health, and clinical practice implications, which we believe can justify the conclusions.

Comments on the Quality of English Language

The entire manuscript should be carefully proofread to ensure proper English and scientific writing.

All our papers, as well as the current text, have been reviewed and corrected by Meryl Jons, a professional medical manuscript translator at the WYN Academy in Mallorca. We trust that all defects have been corrected.

Dear reviewer, we have proceeded to answer all the questions raised and have made changes to the manuscript. To facilitate their location in the paper, we have highlighted the modifications in red.

We greatly appreciate your reflections and advice to improve our article. We believe that, following your recommendations, the contribution of our work to scientific knowledge has been enhanced and clarified. We trust that we have addressed all your questions and remain at your disposal for any further clarification.

Thank you very much.

Round 2

Reviewer 1 Report

Comments and Suggestions for Authors

The study is descriptive and has many problems, such as Ramadan is associated with intermittent fasting. Ramadan is a religious situation that people have adopted all their lives. Intermittent fasting is a protocol that induces physiological adaptations in the body. I insist on the comments in my first review and don´t have any more questions.

Author Response

The study is descriptive and has many problems, such as Ramadan is associated with intermittent fasting. Ramadan is a religious situation that people have adopted all their lives. Intermittent fasting is a protocol that induces physiological adaptations in the body. I insist on the comments in my first review and don´t have any more questions.

We agree with you that Ramadan is a religious celebration that individuals have adopted throughout their lives, and we respect the religious and social beliefs of everyone. Furthermore, we would like to add that, as highlighted in our paper, there is no single diet that is universally beneficial for health. The Mediterranean diet meets the characteristics of the consensus on healthy diets recommended by WHO and FAO (https://www.who.int/publications/i/item/9789240101876#:~:text=Healthy%20diets%20promote%20health%2C%20growth,foodborne%20diseases%20and%20promote%20wellbeing. (Accessed on October 28, 2024).

Additionally, we are not aware of any scientific evidence demonstrating benefits or harms to health from consuming one or more meals per day in healthy individuals.

The Mediterranean diet corresponds to that of the countries in the Mediterranean basin, many of which have a Muslim population that has historically celebrated their religious festivals, including Ramadan. To the best of our knowledge, Ramadan recommends fasting during the day and allows for food intake when night falls. Furthermore, this celebration permits individuals who are ill (e.g., those with diabetes) to abstain from following these precepts.

In our study, we evaluated adherence to the Mediterranean diet using the PREDIMED questionnaire, which is validated for such studies and does not reference the number of meals consumed daily in any of its questions.

Regarding your previous comments, we believed we had adequately addressed your questions. Just as we utilized the PREDIMED questionnaire to assess adherence to the Mediterranean diet, we also employed the IPAQ questionnaire to evaluate the level of physical activity. This questionnaire is internationally recognized and validated for such studies.

We trust that the answers will be to your liking and we appreciate all the recommendations made to improve our manuscript.

Reviewer 4 Report

Comments and Suggestions for Authors

This reviewer would like to express sincere appreciation to the authors for their efforts in improving the manuscript. The revisions made thus far have improved the overall quality of the study. However, several important concerns remain that should be addressed to further strengthen the manuscript before it can be considered for further review.

Major issues:

1. The term ALLY is introduced in Line 61 as a useful measure, but there is no explanation of what it represents in practical terms (i.e., how ALLY is calculated, its assumptions, or its significance). Furthermore, the calculation method for ALLY is missing in the Materials and Methods section, which is problematic since it is a key measure used in the analysis. The results (Tables 2 and 3) present ALLY values, but without a proper explanation of its meaning or calculation method, interpreting these results becomes difficult for the reader. Additionally, there is no explanation of how the threshold for high ALLY metabolic age was set or whether this is a standardized measure. For example, the paper discusses 12 years above chronological age as a cutoff for metabolic age (L25), but does not explicitly connect this with the ALLY metric. More context should also be provided in the discussion to explain the implications of ALLY and how it compares to other health indicators related to metabolic age.

2. The statistical approach described in the revised manuscript raises several critical points that need to be addressed:

a) Multinomial logistic regression is typically used when the dependent variable has more than two categories, which would be appropriate if high ALLY Metabolic Age is categorized into multiple distinct groups. However, the paper does not clarify whether the dependent variable was indeed divided into more than two categories. The authors need to justify the use of multinomial logistic regression by clearly specifying how high ALLY Metabolic Age was categorized.

b) While stratification by age can control for age-related effects, it does not account for other confounding variables that could impact the relationship between the independent variables and metabolic age. The authors should clarify whether they included other confounding factors (e.g., comorbidities, medication use) in the model to ensure a more robust analysis.

c) The authors claim that their approach of evaluating each variable "independently" eliminated confounding factors (L241-242). While this reviewer is not a statistics expert, this statement appears misleading. Logistic regression can adjust for multiple variables simultaneously, but it does not eliminate confounding unless all potential confounders are properly identified and included in the model. The authors should provide a clearer explanation of how the model ensures that confounding was adequately controlled.

Other important Issues:

1. Title: The revised title is an improvement, as it now mentions specific variables (biological age, sex, social class, lifestyle habits). However, it could still be more precise and informative by clearly reflecting the key findings and study population. For example: Associations Between Metabolic Age, Sociodemographic Variables, and Lifestyle Factors in Spanish Workers

2. Introduction: Please make the Introduction more concise, limiting it to three paragraphs (excluding the statement of purpose), with clearer and more focused text to improve flow and readability. You may consider the following:

a) remove redundant explanations of metabolic age;  

b) combine discussions of lifestyle factors (diet, physical activity, stress, and sleep) into a streamlined paragraph;

c) place more emphasis on the relevance of the study to the specific population being examined; and

d) limit general background information and transition directly into the research rationale.

3. Results: a) The table titles and footnotes in the revised paper are not presented in a clear or standardized format. The authors should revise the table titles to ensure they clearly and concisely reflect the content. Footnotes should be added or improved to provide definitions and explanations for any abbreviations, statistical terms, or variables used in the tables to ensure clarity for the reader.

b) Terms related to variables, such as physical activity, are used inconsistently across different tables (e.g., Table 1 vs. Table 2). Consistency in terminology should be ensured across all tables.

c) The text in the Results section should better describe the data presented in the tables and provide a stronger connection to the study’s objectives. Additionally, the authors should clearly indicate relevant statistical tests and significance levels, in the results narrative or in the table footnotes as appropriate.

4. Discussion: a) Some statements are oversimplified and lack sufficient scientific depth. They fail to properly contextualize findings within existing literature. For example, “The second variable studied in our work with the greatest influence on metabolic age is diet,” (L333-334). It does not fully capture the complexity of the study's findings or provide adequate scientific evidence to support the role of diet. Instead of merely stating that diet has the second greatest influence, the authors should explain how adherence to the Mediterranean diet affects metabolic age and connect this to established scientific research. Doing so would enhance the credibility of the findings and better align the discussion with broader dietary research. References to relevant scientific studies would strengthen the analysis and align it with broader research.

b) Please remove redundant explanations and consolidate repetitive sections (e.g., on sex differences, aging, and socioeconomic status) to avoid unnecessary repetition. Additionally, focus on presenting new insights rather than restating findings. Emphasize novel interpretations or comparisons with the literature. Rephrase any sentences that are repeated or echoed in both the discussion and conclusion sections to ensure clarity and avoid redundancy.

5. Scientific writing: The scientific writing in this revised version remains a matter of concern. Some typical examples (this list is not exhaustive) include:

L26-27: 'The variables associated with the most significant risk factors were identified using multivariate methods.' This sentence is vague and lacks detail.

L33-36: 'Metabolic age is influenced by a combination of demographic and ... low adherence to the Mediterranean diet are all factors that can increase metabolic age.' The phrase 'can increase metabolic age' does not appropriately express the study's findings. It could be more accurately stated as: '...Mediterranean diet being associated with higher metabolic age.'

L63-64: 'Metabolic age is influenced by a combination of genetic, physiological, and environmental factors.' This sentence is too general and lacks specificity. It can be revised as: ‘Metabolic age is influenced by a combination of factors, including genetic predisposition, body composition, physical activity, and diet (Cite reference/s).'

L105–106: ‘The sample consisted of employees who underwent occupational health examinations in our service.' This statement lacks clarity regarding the nature of the sample. It does not explain what ‘our service’ refers to or how this might impact the representativeness of the sample.

L187-189: 'Student's t-test for independent samples and the chi-square test for independent samples were used to analyse bivariate associations (to compare means).' While the Student's t-test is appropriate for comparing means between two groups, the chi-square test is not used to compare means. Instead, it is used to test associations between categorical variables (e.g., frequencies or proportions, not means).

L211-212: 'The obtained results generally favor women regarding these mean values.' This sentence is imprecise and vague. It should be clarified to more accurately reflect the findings.

L281-283: 'Women, especially after menopause, experience hormonal changes that can increase abdominal fat and decrease muscle mass, contributing to an increase in metabolic age.' Although this statement is accurate, it lacks scientific depth. A more precise version would be: 'Women, especially after menopause, undergo hormonal changes, such as reduced estrogen levels, which can lead to increased abdominal fat and decreased muscle mass, contributing to a higher metabolic age.

6. References/Citations: In the revised text, Roman numerals have been used for reference citations. But in the References section, papers are cited using both Arabic and Roman numerals. Please ensure that all citations are switched to Arabic numerals, as Nutrients typically uses Arabic numerals for referencing.

Comments on the Quality of English Language

The English in this manuscript is understandable; however, the scientific writing remains a concern and requires significant improvement.

Author Response

This reviewer would like to express sincere appreciation to the authors for their efforts in improving the manuscript. The revisions made thus far have improved the overall quality of the study. However, several important concerns remain that should be addressed to further strengthen the manuscript before it can be considered for further review.

First, we would like to thank the reviewer for recognizing the effort made by the authors. Secondly, we appreciate your new suggestions to improve the article.

Below, we respond to each of your recommendations.

Major issues:

  1. The term ALLY is introduced in Line 61 as a useful measure, but there is no explanation of what it represents in practical terms (i.e., how ALLY is calculated, its assumptions, or its significance). Furthermore, the calculation method for ALLY is missing in the Materials and Methods section, which is problematic since it is a key measure used in the analysis. The results (Tables 2 and 3) present ALLY values, but without a proper explanation of its meaning or calculation method, interpreting these results becomes difficult for the reader. Additionally, there is no explanation of how the threshold for high ALLY metabolic age was set or whether this is a standardized measure. For example, the paper discusses 12 years above chronological age as a cutoff for metabolic age (L25), but does not explicitly connect this with the ALLY metric. More context should also be provided in the discussion to explain the implications of ALLY and how it compares to other health indicators related to metabolic age.

We have proceeded in the material and methods section to define ALLY, specify its calculation and the reason for the cut-off point at 12 years.

“ALLY (Avoidable Lost Life Years) were calculated by subtracting metabolic age from biological age. Some published studies have reported that a difference of at least 12 years between chronological and metabolic age reduces cardiovascular risk. ALLY is classified as low if the difference is less than 3 years, normal if it is between 3 and 11 years, and high if the difference is 12 years or more. A metabolic age 12 years or more above the chronological age was considered high[ ]. This serves as the cutoff point for establishing high metabolic age values for ALLY (Avoidable Lost Life Years).”

  1. The statistical approach described in the revised manuscript raises several critical points that need to be addressed:
  2. a) Multinomial logistic regression is typically used when the dependent variable has more than two categories, which would be appropriate if high ALLY Metabolic Age is categorized into multiple distinct groups. However, the paper does not clarify whether the dependent variable was indeed divided into more than two categories. The authors need to justify the use of multinomial logistic regression by clearly specifying how high ALLY Metabolic Age was categorized.

ALLY is classified as low if it is less than 3 years, normal if it falls between 3-11 years, and high if it exceeds 12 years. Therefore, since the dependent variable has more than two categories, we applied multinomial logistic regression.

  1. b) While stratification by age can control for age-related effects, it does not account for other confounding variables that could impact the relationship between the independent variables and metabolic age. The authors should clarify whether they included other confounding factors (e.g., comorbidities, medication use) in the model to ensure a more robust analysis.

Confounding factors, as you rightly pointed out, were not included because they were not considered statistically or biologically plausible. Thank you very much for your observation.

  1. c) The authors claim that their approach of evaluating each variable "independently" eliminated confounding factors (L241-242). While this reviewer is not a statistics expert, this statement appears misleading. Logistic regression can adjust for multiple variables simultaneously, but it does not eliminate confounding unless all potential confounders are properly identified and included in the model. The authors should provide a clearer explanation of how the model ensures that confounding was adequately controlled.

You are absolutely correct; we have removed the expression "eliminated confounding factors."

While recognizing that other potential influencing factors may exist, the authors have included in the model those factors deemed biologically and statistically plausible. 

Other important Issues:

  1. Title: The revised title is an improvement, as it now mentions specific variables (biological age, sex, social class, lifestyle habits). However, it could still be more precise and informative by clearly reflecting the key findings and study population. For example: Associations Between Metabolic Age, Sociodemographic Variables, and Lifestyle Factors in Spanish Workers

We appreciate your suggestion and have modified the title as you suggested. Thank you very much.

  1. Introduction:Please make the Introduction more concise, limiting it to three paragraphs (excluding the statement of purpose), with clearer and more focused text to improve flow and readability. You may consider the following:
  2. a) remove redundant explanations of metabolic age;  
  3. b) combine discussions of lifestyle factors (diet, physical activity, stress, and sleep) into a streamlined paragraph;
  4. c) place more emphasis on the relevance of the study to the specific population being examined; and
  5. d) limit general background information and transition directly into the research rationale.

Following your suggestions, we have modified the introduction, making it more concise and improving the text.

  1. Results:a) The table titles and footnotes in the revised paper are not presented in a clear or standardized format. The authors should revise the table titles to ensure they clearly and concisely reflect the content. Footnotes should be added or improved to provide definitions and explanations for any abbreviations, statistical terms, or variables used in the tables to ensure clarity for the reader.

Table titles, footnotes, abbreviations and statistical terms used have been reviewed.

  1. b) Terms related to variables, such as physical activity, are used inconsistently across different tables (e.g., Table 1 vs. Table 2). Consistency in terminology should be ensured across all tables.

It has been homogenized in all tables. Thank you for your observation.

  1. c) The text in the Results section should better describe the data presented in the tables and provide a stronger connection to the study’s objectives. Additionally, the authors should clearly indicate relevant statistical tests and significance levels, in the results narrative or in the table footnotes as appropriate.

The statistical tests performed on each table and their significance levels have been indicated. Thank you very much.

  1. Discussion:a) Some statements are oversimplified and lack sufficient scientific depth. They fail to properly contextualize findings within existing literature. For example, “The second variable studied in our work with the greatest influence on metabolic age is diet,” (L333-334). It does not fully capture the complexity of the study's findings or provide adequate scientific evidence to support the role of diet. Instead of merely stating that diet has the second greatest influence, the authors should explain how adherence to the Mediterranean diet affects metabolic age and connect this to established scientific research. Doing so would enhance the credibility of the findings and better align the discussion with broader dietary research. References to relevant scientific studies would strengthen the analysis and align it with broader research.

We have revised the wording to clarify and assert the results more effectively, drawing comparisons with various related studies, as to our knowledge, this is the first study to establish associations between the studied variables and metabolic age.

  1. b) Please remove redundant explanations and consolidate repetitive sections (e.g., on sex differences, aging, and socioeconomic status) to avoid unnecessary repetition. Additionally, focus on presenting new insights rather than restating findings. Emphasize novel interpretations or comparisons with the literature. Rephrase any sentences that are repeated or echoed in both the discussion and conclusion sections to ensure clarity and avoid redundancy.

We have revised the wording to clarify and assert the results more effectively, drawing comparisons with various related studies, as to our knowledge, this is the first study to establish associations between the studied variables and metabolic age.

  1. Scientific writing:The scientific writing in this revised version remains a matter of concern. Some typical examples (this list is not exhaustive) include:

L26-27: 'The variables associated with the most significant risk factors were identified using multivariate methods.' This sentence is vague and lacks detail.

We have revised the statement as follows:

"Using multivariate models, specifically multinomial logistic regression, we observed that all independent variables (sex, age, socioeconomic class, physical activity, adherence to the Mediterranean diet, and smoking) displayed varying levels of association with elevated metabolic age values. Among these independent variables, those showing the highest degree of association, represented by odds ratios, were physical activity, adherence to the Mediterranean diet, and socioeconomic class. In all cases, the observed differences demonstrated a high level of statistical significance (p<0.001)."

L33-36: 'Metabolic age is influenced by a combination of demographic and ... low adherence to the Mediterranean diet are all factors that can increase metabolic age.' The phrase 'can increase metabolic age' does not appropriately express the study's findings. It could be more accurately stated as: '...Mediterranean diet being associated with higher metabolic age.'

We have proceeded to modify the sentence as suggested. Thank you very much.

L63-64: 'Metabolic age is influenced by a combination of genetic, physiological, and environmental factors.' This sentence is too general and lacks specificity. It can be revised as: ‘Metabolic age is influenced by a combination of factors, including genetic predisposition, body composition, physical activity, and diet (Cite reference/s).'

We have proceeded to modify the sentence as suggested. Thank you very much.

L105–106: ‘The sample consisted of employees who underwent occupational health examinations in our service.' This statement lacks clarity regarding the nature of the sample. It does not explain what ‘our service’ refers to or how this might impact the representativeness of the sample.

We have modified the sentence to make the sample more clear. Thank you very much.

L187-189: 'Student's t-test for independent samples and the chi-square test for independent samples were used to analyse bivariate associations (to compare means).' While the Student's t-test is appropriate for comparing means between two groups, the chi-square test is not used to compare means. Instead, it is used to test associations between categorical variables (e.g., frequencies or proportions, not means).

It was a writing error. The T-student test was used to compare means and the chi-square test was used to compare proportions.

We have corrected it. Thank you very much for your warning

L211-212: 'The obtained results generally favor women regarding these mean values.' This sentence is imprecise and vague. It should be clarified to more accurately reflect the findings.

We have modified the sentence, clarifying the findings.

“In nearly all values obtained, the mean metabolic age is lower in women compared to men.”

L281-283: 'Women, especially after menopause, experience hormonal changes that can increase abdominal fat and decrease muscle mass, contributing to an increase in metabolic age.' Although this statement is accurate, it lacks scientific depth. A more precise version would be: 'Women, especially after menopause, undergo hormonal changes, such as reduced estrogen levels, which can lead to increased abdominal fat and decreased muscle mass, contributing to a higher metabolic age.

We have proceeded to modify the sentence as suggested. Thank you very much. 

  1. References/Citations: In the revised text, Roman numerals have been used for reference citations. But in the References section, papers are cited using both Arabic and Roman numerals. Please ensure that all citations are switched to Arabic numerals, as Nutrientstypically uses Arabic numerals for referencing.

Modified and placed in brackets, thanks.

Dear reviewer, we have proceeded to answer all the questions raised and have made changes to the manuscript. To facilitate their location in the paper, we have highlighted the modifications in red.

We trust that the answers will be to your liking and we appreciate all the recommendations made to improve our manuscript.
